# Climatic Change and Habitat Availability for Three Sotol Species in México: A Vision towards Their Sustainable Use

Jorge Luis Becerra-López [1,*], Rigoberto Rosales-Serna [2], Muhammad Ehsan [3],
Jesús Salvador Becerra-López [4], Alexander Czaja [5], José Luis Estrada-Rodríguez [5],
Ulises Romero-Méndez [5], Saúl Santana-Espinosa [2], César Manuel Reyes-Rodríguez [2],
Julio César Ríos-Saucedo [2] and Pablo Alfredo Domínguez-Martínez [2]

[1] Laboratorio de Cambio Climático y Conservación de Recursos Naturales, Facultad de Ciencias Biológicas, Universidad Juárez del Estado de Durango, Av. Universidad S/N Fracc. Filadelfia, Gómez Palacio, Dgo C. P. 35020, Mexico
[2] INIFAP-Durango. Campo Experimental Valle del Guadiana, Carr. Durango-El Mezquital km 4.5, Durango, Dgo C. P. 34170, Mexico; rosales.rigoberto@inifap.gob.mx (R.R.-S.); santana.saul@inifap.gob.mx (S.S.-E.); ceourreyrod@gmail.com (C.M.R.-R.); rios.julio@inifap.gob.mx (J.C.R.-S.); dominguez.pablo@inifap.gob.mx (P.A.D.-M.)
[3] Centro de Bachillerato Tecnológico Agropecuario 162. Carr. Mexico-Veracruz vía Texcoco km 95, Francisco I. Madero, Tlax C. P. 90280, Mexico; ehsanm2000@hotmail.com
[4] Laboratorio de Química y Física de Suelos, Facultad de Agricultura y Zootecnia, Universidad Juárez del Estado de Durango, Carretera Gómez Palacio—Tlahualilo km. 32. Venecia, Gómez Palacio, Dgo C. P. 35111, Mexico; biol.becerrajesus@hotmail.com
[5] Laboratorio de Ecología y Evolución, Centro de Estudios Ecológicos, Facultad de Ciencias Biológicas, Universidad Juárez del Estado de Durango, Avenida Universidad S/N Fracc. Filadelfia, Gómez Palacio, Dgo C. P. 35010, Mexico; aaczaja@hotmail.com (A.C.); josefo7@hotmail.com (J.L.E.-R.); biologo_ulisesromero@yahoo.com.mx (U.R.-M.)
[*] Correspondence: biologo.jlbl@gmail.com

**Abstract:** The industrial production and commercialization of distilled beverages from Sotol plants (Family: *Asparagaceae*, Subfamily: *Zolinoideae* and Genus: *Dasylirion*) has witnessed a steady growth in recent decades; this condition involves a greater use of the raw material that comes almost exclusively from natural populations, which could compromise the sustainability of the marginalized areas of Northern México. In the present work, habitat availability was evaluated for the presence and use of the species; *Dasylirion wheeleri* (S. Watson ex Rothr.), *Dasylirion cedrosanum* (Trelease), and *Dasylirion durangense* (Trelease) in México, considering different scenarios of climate change. For this purpose, we used the niche's spatial analysis from a correlative approach. The results show that under the current climate, the species studied have a predominantly low habitat suitability in the central region of the Chihuahuan desert, with *D. cedrosanum* being the species with the lowest suitability. For the year 2050, a slight increase in the habitat suitability is expected only for *D. durangense* and *D. wheeleri*. This suggests that *D. cedrosanum* could be considered as the species with the greatest vulnerability to the possible climatic variations generated by global climate change. Likewise, the analysis of the results indicates that *D. durangense* is the one who has the greatest potential for use; however, it is necessary to consider that many of its populations may be climatically stressed and an inappropriate use could put them at risk.

**Keywords:** *Dasylirion*; climatic change; niche; sustainable

## 1. Introduction

Sotol is a plant found in the arid regions of México; it grows abundantly in the states of Chihuahua, Durango, Coahuila, and in many other regions of the Chihuahuan Desert, being of ecological and cultural importance in this region [1]. For example, from an ecological aspect, it has been reported that natural Sotol populations perform an important ecological function due to dry biomass production, which contributes to organic matter accumulation in shallow and rocky soils on the hills and the mountain sides where this species grows [2].

From a cultural point of view, it has been documented that the Sotol plants were used by the early civilizations that settled in the Northern part of México, among which were the irritilas and guachichiles who used the plant as a source of food during religious ceremonies called mitotes [3]. Likewise, for several decades now, for the inhabitants of arid and semi-arid zones, Sotol species constitute a natural resource of great value, since they are used as fodder for livestock [4], fence construction, basket making [5], and the production and marketing of distilled beverages from Sotol. It provides an economic resource to the inhabitants of marginalized areas of Northern México (States of Chihuahua, Durango and Coahuila) [6].

Currently, the production and marketing of distilled beverages from Sotol is being carried out in an industrial manner, in which the natural populations of various *Dasylirion* plant species are used to meet the industry's requirements [7]. According to reported statistics, the production of alcoholic beverages from *Dasylirion* species show growth; for example, the income in México from sales of Sotol in 2005 was around 19.5 million, while for 2008, 58.2 million pesos was registered [8]. This growth implies the need for more raw materials derived almost exclusively from natural populations, which has caused a drastic decrease in *Dasylirion* populations in some states such as Chihuahua and Durango [6].

Keeping in view the fact that these species are distributed mostly in arid and semi-arid ecosystems [9], this may be considered as a fragile environment. In this regard, producers and the Mexican Council of Sotol, AC, have been driven by the interest of ensuring the sustainability of the industry. Their preference is establishing priority areas for the reforestation of different species of Sotol to ensure that ecosystems are not negatively affected. Moreover, it guarantees the smooth supply of raw materials from natural populations [8].

In this way, the suitability of a species' habitat and its vulnerability can be assessed through the ecological niche modeling [10,11]. In recent years, tools have emerged that enable the ecological niche of species to be modeled and projected across geographic space to identify potentially suitable areas for a given species [12]. Furthermore, these correlative models, based on the ecological niche, allow the estimation of the possible impact of climate change on the habitat suitability of the species [13]. In this regard, MaxEnt has demonstrated good predictability based on presence data [14,15]. The model is based on the statistical principle of maximum entropy that facilitates making predictions with the use of incomplete information, which represents an advantage, since for most taxa there is a lack of data regarding true absences [16].

The decrease witnessed in the populations of Sotol Species in Northern México undermines the ecosystem stability and consequently the socioeconomic stability of the region. The present study, therefore, aims at identifying potential areas for *Dasylirion* species; *D. wheeleri*, *D. cedrosanum*, and *D. durangense* in México, as well as assessing the vulnerability of these areas to climate change. This information will identify potential sites for the establishment of Sotol producing units, as well as the priority species for this use.

## 2. Methodology

### 2.1. Presence Data

Occurrence data were compiled from Global Biodiversity Information Facility database (GBIF, http://www.gbif.org). Through a filtration process, for each species, all those erroneous and unreliable geographical records that did not match the known distribution of the species under study

were deleted. For this, the information provided by the National Institute of Forest, Agricultural and Livestock Research was taken as a reference [17], obtaining a total of 107 geographic records for *Dasylirion cedrosanum*, 25 for *Dasylirion durangense*, and 136 *Dasylirion wheeleri* (Figure 1).

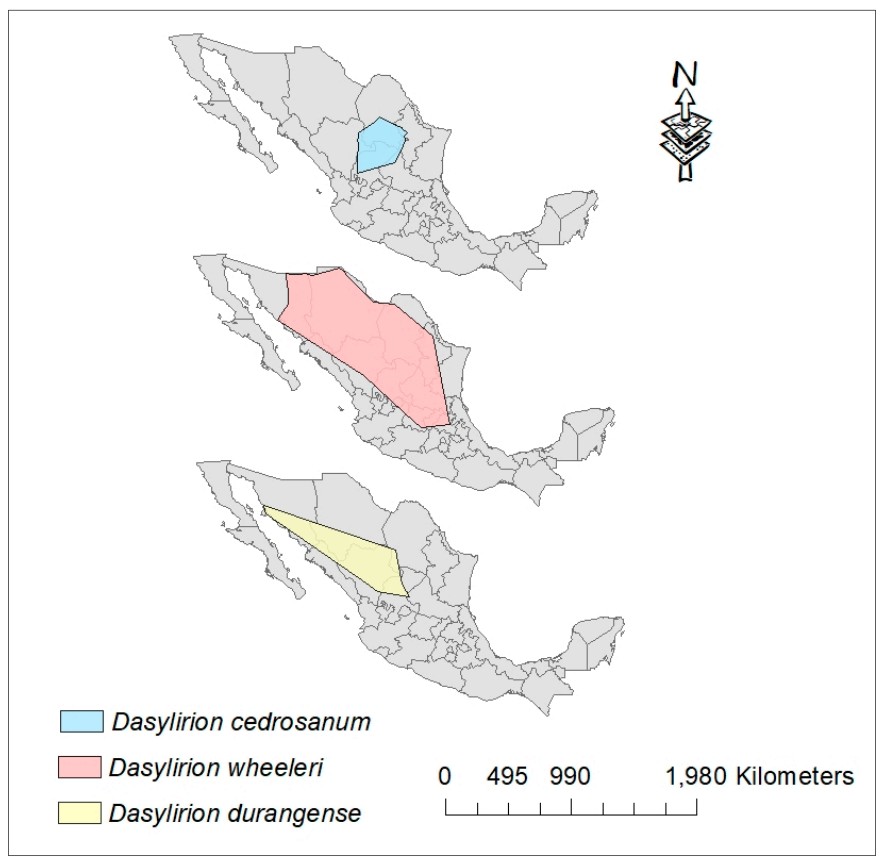

**Figure 1.** The geographical range in which the species under study can be found is presented.

### 2.2. Climatic Variables

Climatic information was obtained from 19 current climatic layers and elevation data, available in WorldClim database version 1.4 [18], and the variables (19 in total) are: annual mean temperature (bio1), mean diurnal temperature range [mean of monthly (max temp - min temp)] (bio2), isothermality (bio3), temperature seasonality (bio4), max temperature of warmest month (bio5), min temperature of coldest month (bio6), temperature annual range (bio7), mean temperature of wettest quarter (bio8), mean temperature of driest quarter (bio9), mean temperature of warmest quarter (bio10), mean temperature of coldest quarter (bio11), annual precipitation (bio12), precipitation of wettest month (bio13), precipitation of driest month (bio14), precipitation seasonality (bio15), precipitation of wettest quarter (bio16), precipitation of driest quarter (bio17), precipitation of warmest quarter (bio18), and precipitation of coldest quarter (bio19). These layers contain climatic averages of weather conditions recorded from 1950–2000 with a spatial resolution of 30 arc-seconds (~1 km).

For the selection of environmental variables, 10,000 background points were added to the polygon distribution area of *D. cedrosanum*, *D. durangense*, and *D. wheeleri*. The information of the 19 environmental variables from the current climate was added to these points. With the generated information, a bivariate correlation analysis was conducted in order to reduce the multicollinearity between the input variables [12,13]; predictor variables that were highly correlated ($|r| \geq 0.7$) were excluded [19]. The addition of 10,000 random points was based on the criteria of not discriminating (nonrepetitive) relevant information but segregated geographical areas with important climatic information within the range of the species under study [20]. The climatic variables selected were:

annual mean temperature, mean diurnal temperature range, temperature seasonality, temperature annual range, annual precipitation, precipitation of driest month, and precipitation seasonality.

### 2.3. Climate Profile and Niche Range

The input variables were used to perform a principal component analysis (PCA) in the R software (R version 3.1.3, library ecospat) [21,22]. From the information shown in the PCA, the climate profile of the study area was identified and the distribution of the species in the climatic environment was evaluated (niche range).

### 2.4. Niche Modeling

A maximum entropy model (MaxEnt, version 3.3.3k) [16,23] was used to represent the potential habitat suitability of *D. cedrosanum*, *D. durangense*, and *D. wheeleri*. MaxEnt predicts habitat suitability as a function of environmental variables and species occurrence data. This habitat suitability is represented by a scale ranging from 0 (low suitability) to 1 (high fitness) [24–27]. Proper calibration and evaluation are necessary to reduce the complexity of the model [28], considering the choice of: (i) accessible area (background or M area), (ii) the type of variables that MaxEnt constructs (features), and (iii) the type of model output (raw, cumulative, or logistic), as these considerations affect the inferences to be made [29].

In this study, the calibration and evaluation method for *D. cedrosanum*, *D. durangense*, and *D. wheeleri* modeling were carried out using the library "ENMeval" [30] in the statistical software R 3.1.3 [21]. The calibrated model was evaluated by calculating the coefficient standardized Akaike information criterion (AICc). The AICc provides information on the relative quality of a model [28]. As the AICc is calculated using the data set, it is not affected by the method chosen for the data partition [30]. The model with the lowest AICc was selected as the best fit for the species.

The information obtained from the calibrated models was projected to México using the software MaxEnt [16] considering the environmental variables described above and the soil layers of litosol and calcareous regosol, as well as the elevation. There were 100 repetitions performed [31] to obtain an ecological niche model, geographically represented as a map of habitat suitability under current climatic conditions for *D. cedrosanum*, *D. durangense*, and *D. wheeleri*. Likewise, the information obtained from the calibrated models was projected to México considering soil layers of litosol, calcareous regosol, elevation, and the climate scenario proposed by Beijing Climate Center Climate System Model (BCC-CSM1-1) for the year 2050 under the concentration of greenhouse gases RCP 2.6 = +2.6 W/m$^2$ and RCP 8.5 = +8.5 W/m$^2$. For the validation of the models, the value of AUC (Area under the ROC curve) was considered.

The importance of each bioclimatic variable in the observed distribution of *D. cedrosanum*, *D. durangense* and *D. wheeleri* was evaluated according to the relative importance of each variable, which was obtained by adding the percentage of contribution (PC) and the importance of permutation (IP), evaluated by MaxEnt, and the result was divided by two (average contribution (PC + IP)/2) [32].

## 3. Results

### 3.1. Climate Profile and Niche Range

The principal component analysis indicates that component one explains 94% of variation in the data with a standard deviation of: 572.9, in the climate context, within the study area. While component two only explains 5.9% of this variation, standard deviation: 141.6. The influence graph indicates that the variable annual precipitation has a greater positive association with component one, while variable temperature seasonality shows a greater negative relationship with component two (Figure 2).

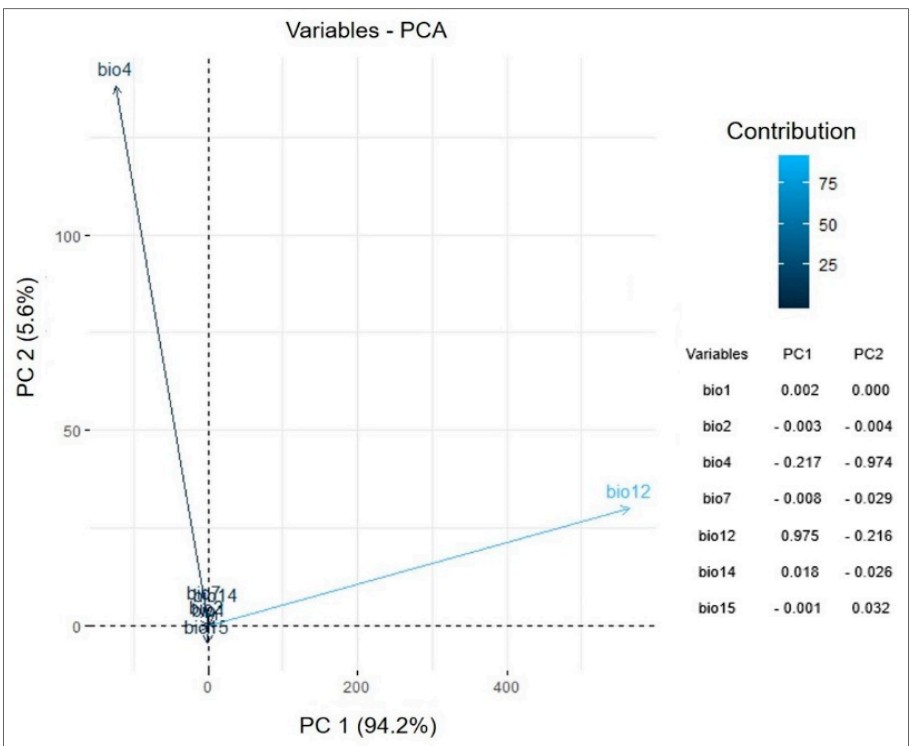

**Figure 2.** Contribution of the climatic variables in the grouping of components showing that the variables annual precipitation (bio12) and temperature seasonality (bio4) are those that have the greatest weight in the climatic profile of the study area.

Regarding the size of the niches of the species under study, our results indicate that *D. cedrosanum* is the one with the lowest amplitude ranges for the annual precipitation and temperature seasonality variables, *D. durangense* showed the highest amplitude for the annual precipitation variable, while *D. wheeleri* is the one with the highest amplitude for the temperature seasonality variable (Figure 3 and Table 1).

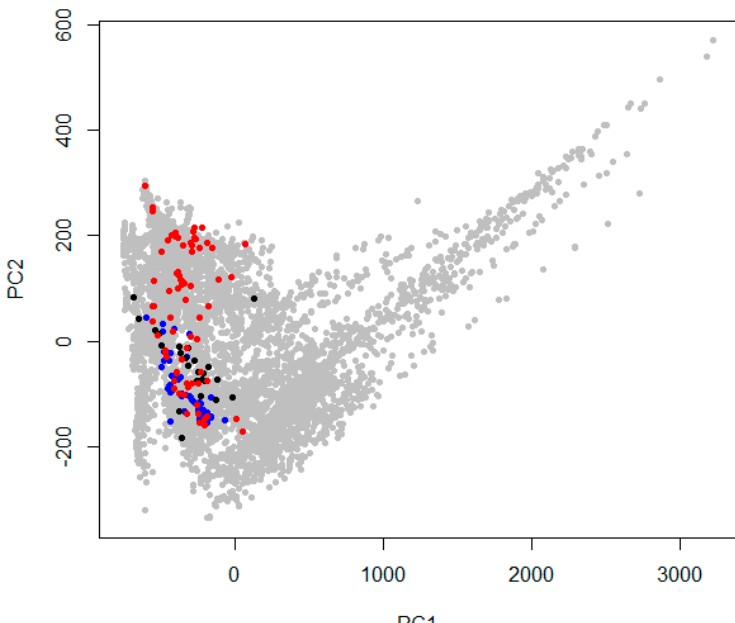

**Figure 3.** Distribution of the species under study in the climate context; *D. cedrosanum* (blue dots), *D. durangense* (black dots), and *D. wheeleri* (red dots).

**Table 1.** The ranges of species distribution are shown in the climatic context. It can be seen that *D. cedrosanum* is the one with the lowest occupation ranges.

| Species | Component 1 | Component 2 |
|---|---|---|
| *D. cedrosanum* | −596.06939−−70.51727 | −152.90965−67.29285 |
| *D. durangense* | −680.8457−130.0315 | −182.99698−83.10202 |
| *D. wheeleri* | −612.12476−67.81533 | −170.2067−295.7655 |

*3.2. Niche Modeling*

Out of the calibrations of the models for the species *D. cedrosanum*, *D. durangense*, and *D. wheeleri*, the one that obtained a value of 0 was selected. This was under the criterion that the candidate calibrations must be classified according to their AICc, and the calibration that presents the minimum AICc will be the best. On the other hand, the models obtained for the species *D. cedrosanum*, *D. durangense*, and *D. wheeleri* showed an AUC of 0.94, 0.80, and 0.83, respectively; indicating low levels of commission (predicts the presence of the species where it does not exist, false positive) and omission (predicts the non presence of the species where it really exists, false negative). The contribution analysis indicates that the precipitation of the driest month variable provides the most information to explain the habitat suitability of *D. cedrosanum* and *D. durangense* species. For *D. wheeleri*, the annual mean temperature variable provided the most information to explain its habitat suitability (Table 2). Likewise, Jacknife's analysis indicates that, when working on the model using the variables individually, for the species *D. cedrosanum* and *D. durangense*, the performance of the model is higher when working only with the precipitation of the driest month variable and this yield decreases to a greater extent when this variable is not considered in the model. For *D. wheeleri*, the performance of the model is greater when working individually with the annual mean temperature and annual precipitation variable, and this yield decreases to a greater extent when this last variable is not considered in the model (Figure 4).

**Table 2.** Relative importance of climatic variables in the habitat suitability of the species under study. The values presented for each variable and species are the average contribution [(PC + PI)/2].

|  | *D. cedrosanum* | *D. durangense* | *D. wheeleri* |
|---|---|---|---|
| Annual Mean Temperature | 0.8 | 6.7 | 34.2 |
| Mean Diurnal Temperature Range | 23.5 | 0.0 | 9.2 |
| Temperature Seasonality | 5.3 | 9.5 | 22.7 |
| Temperature Annual Range | 4.5 | 16.6 | 1.5 |
| Annual Precipitation | 0.8 | 0.0 | 23.8 |
| Precipitation of Driest Month | 58.6 | 59.2 | 8.7 |
| Precipitation Seasonality | 6.6 | 3.2 | 0.1 |

Under current weather conditions, the models indicate that for *D. cedrosanum*, the areas with greater habitat suitability are reduced and are distributed in the northwest and southeast of the state of Coahuila, and the rest of the state has a habitat suitability that goes from low to null. For the north-central region of the state of San Luis Potosí, the model indicates intermediate habitat suitability for this species. While the southeast of the state of Chihuahua, northeast of Durango, north of Zacatecas, and south of Nuevo León and Tamaulipas present a suitability of habitat from low to null. Considering the climate change scenario RCP 8.5, the model indicates habitat suitability similar to that obtained under current weather conditions. While under the RCP scenario 2.6, areas with high habitat suitability decrease with respect to the projections obtained under current climate criteria (Figure 5).

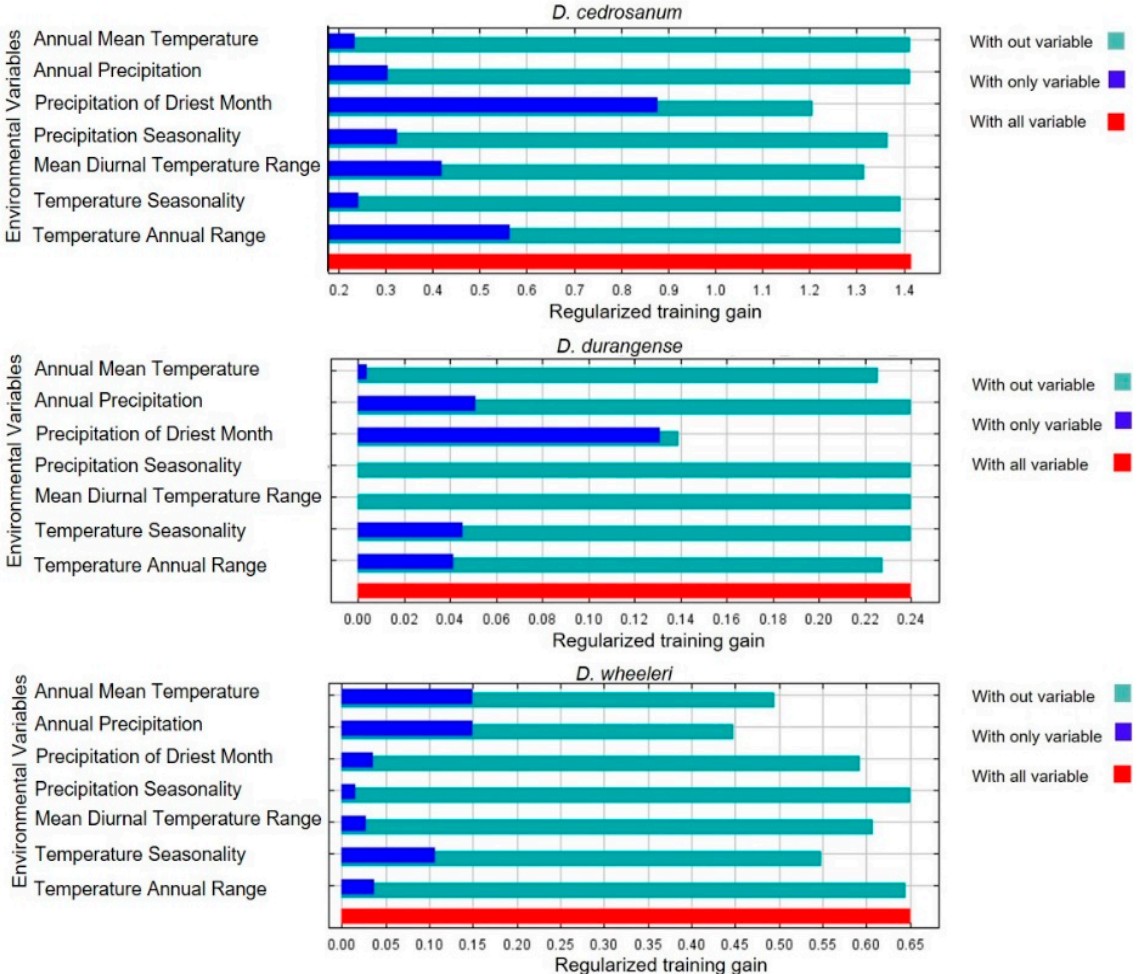

**Figure 4.** The Jackknife test is shown, which reports on the climatic variables that are most important to explain the distribution of the species under study.

For the *D. durangense* species, the model, under current weather conditions, shows a gradient of habitat suitability that goes from intermediate to null. The states that present these characteristics are Aguascalientes, Chihuahua, Durango, and Zacatecas; the states with only low and null habitat suitability are Sonora, Sinaloa, Nuevo León, Tamaulipas, and San Luis Potosí. While, according to the model, the rest of the states of Mexican Republic have a null habitat suitability. Considering the climate change scenario RCP 2.6 and RCP 8.5, the habitat suitability of *D. durangense* ranges from low to medium in some regions of the north and south of the state of Chihuahua, north of Durango, and northwest of Coahuila, and the rest of the Mexican territory has a configuration in the habitat suitability similar to that obtained in the model considering the current climate (Figure 5).

On the other hand, for *D. wheeleri*, under the current climate criteria, there is a habitat suitability that goes from high to null in the states of Chihuahua, Coahuila, Durango, and Sonora. In the states of Sinaloa, Guanajuato, Jalisco, Querétaro, Nuevo León, and Tamaulipas low to null suitability was observed; the rest of the states show null suitability. Considering the climate change scenario RCP 2.6 and RCP 8.5, the suitability of habitat decreases from high to low in the southeast of Chihuahua, northeast of Durango, and northwest of Coahuila. For the northwest and southeast of Zacatecas, central San Luis Potosí, central Aguascalientes, northeast of Jalisco, and north of Guanajuato, the low and medium habitat suitability becomes high. While the south of San Luis Potosí, east of Guanajuato, south of Nuevo León, and central Querétaro move from low to medium habitat suitability (Figure 5).

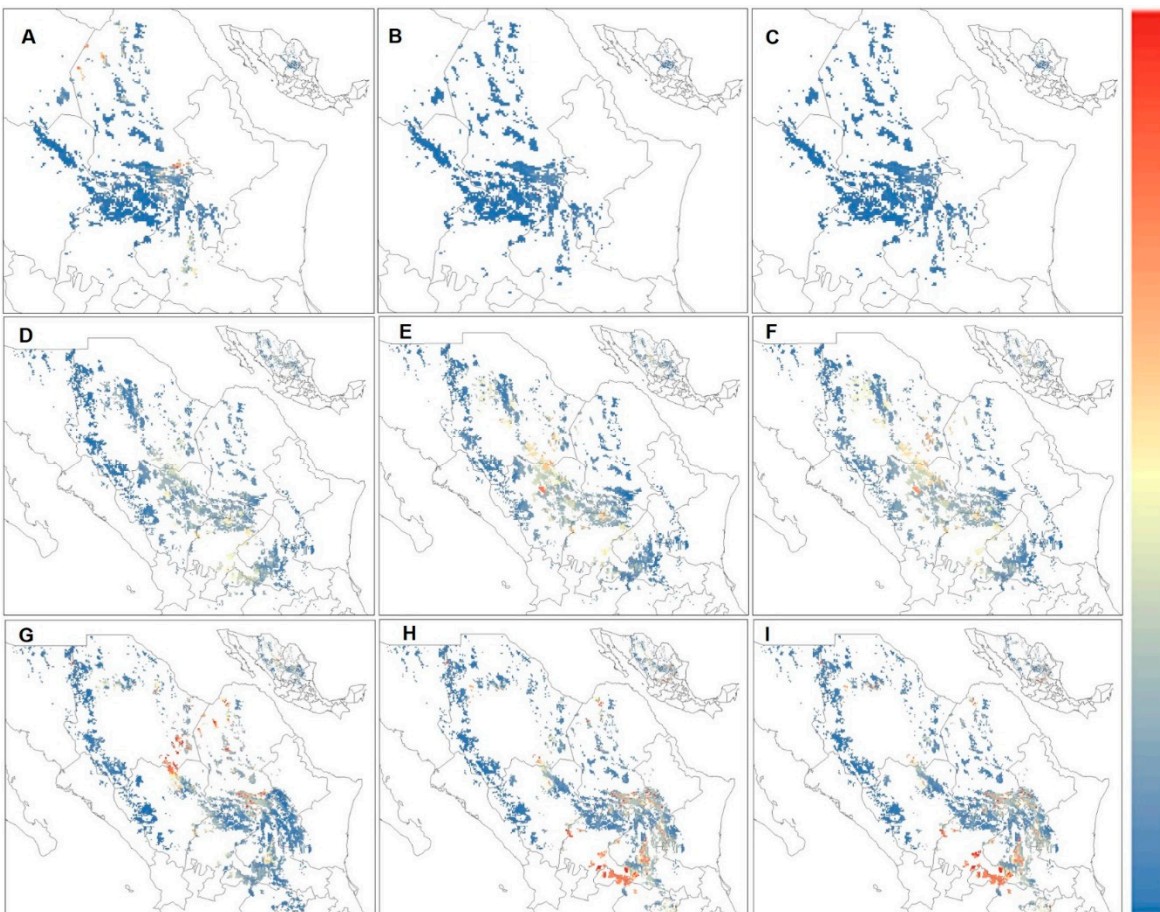

**Figure 5.** Panels (**A**–**C**) indicate habitat suitability for *D. cedrosanum* considering the current climate, scenario RCP 2.6, and RCP 8.5, respectively. Panels (**D**–**F**) correspond to *D. durangense* under current weather conditions, RCP 2.6, and RCP 8.5, respectively. While panels (**G**–**I**) correspond to *D. wheeleri* under current climate criteria, RCP 2.6, and RCP 8.5, respectively. Habitat suitability is presented in a gradient that ranges from high (red) to low (blue), and white areas indicate a zero habitat suitability.

## 4. Discussion

It has been mentioned that each site on earth is characterized by a set of environmental conditions that define a specific habitat inhabited or uninhabited by a community of species [12]. An analysis based on the management of PCA found that the main tendency of variation of flora in arid ecosystems (expressed in the first axis of PCA) corresponds to the rainfall gradient [33]. In this regard, our results indicate that in the study area the climate profile is determined to a greater extent by the annual precipitation variable, followed by the temperature seasonality variable. This suggests that habitat suitability for the species under study may be primarily influenced by variations in the annual precipitation variable.

Various studies suggest that variations in climate generate changes in habitat, which represent one of the greatest threats to biodiversity in the coming decades [34–36], and this threat can be analyzed from the niche perspective [10]. In this regard, it is mentioned that species with broad niches could be less vulnerable to abrupt environmental variations caused by climate change, while species with reduced niches and with restricted distribution ranges may be particularly threatened by climatic alterations [37–39]. For example, within the distribution area of species of the genus *Astrophytum*, a decrease in precipitation and elevation in temperature is projected, so it is expected that those species with the broadest niches could be less affected. This is under the criterion that species with wide niches can occupy different types of environments, which has an important role in its ability to persist in different environmental conditions [40].

Based on the above, it is possible to point out that for our case study, *D. cedrosanum* is the species with the highest vulnerability to climate change, since it shows limited ranges for the variables annual precipitation and temperature seasonality, so it could hardly adapt to possible environmental variations. In this context, *D. wheeleri* and *D. durangense* could present a greater ability to adapt to the climate change variations. Their wide occupation ranges for annual precipitation and temperature seasonality variables suggest the use of a greater number of geographic zones with different climatic environments, which could possibly provide them with a greater capacity to adapt to climate change.

Keeping the above in view, it is expected that due to climate change in the coming decades the average temperature will be noticeably higher, in the deserts of North America [41]. In this sense, it has been mentioned that agaves and cactus thrive under higher temperatures, variable rainfall, and the increase in $CO_2$, characteristic of global climate change [42]. However, it has been reported that under natural conditions, the establishment of populations of different Sotol species is greatly diminished by adverse environmental factors such as drought [43]. In this regard, our results indicate that under the current climate the species under study have a predominantly low habitat suitability with limited areas of medium and high habitat suitability in the central region of the Chihuahuan desert, with *D. cedrosanum* being the species under study with the lowest number of areas having high habitat suitability.

The results obtained in the modeling for the year 2050 under conditions RCP 2.6 and RCP 8.5 show a slight increase in the habitat suitability for *D. durangense* and *D. wheeleri*. However, for *D. cedrosanum*, only low habitat suitability zones are shown. Therefore, it is possible to point out that, although the future climate trend is framed by an increase in $CO_2$ and temperature [44], this condition cannot be correlated in general with an increase in the habitat suitability of the species of the genus *Dasylirion*. Our results reinforce the hypothesis that species with narrow niches are more susceptible to climate change, since in this study the species that presented the smallest climatic niche (*D. cedrosanum*) showed the greatest decrease in the suitability of habitat considering the increase in atmospheric $CO_2$ concentrations. Under the evidence presented, it is possible to point out that the species under study *D. cedrosanum* could be considered as the species with the greatest vulnerability to the possible climatic variations generated by global climate change.

In recent decades, there has been a growing interest in the commercialization of Sotol due, among other factors, to the worldwide popularity of other fermented alcoholic beverages originating in México, such as Tequila and Mezcal [45]. In this regard, the states of Chihuahua, Durango, and Coahuila are the Mexican states with the largest natural population of Sotol and where different species of Sotol have been reported (*Dasylirion* spp.) [46]. However, only the species of *D. cedrosanum* and *D. durangense* have characteristics and properties to be used in the alcohol industry [1,41]. In this respect, our results suggest that despite their large populations, Sotol species are found in stressful climatic environments. Hence, strategies for their industrial use should consider these characteristics, allowing a sustainable use of this nontimber natural resource.

Summarizing the points discussed, our results show that of the two Sotol species of commercial interest mentioned above, *D. cedrosanum* is the one that currently shows a greater climatic vulnerability in its habitat, a condition that appears to continue until 2050. By this measure, *D. durangense* currently presents the greatest potential for utilization, from the viewpoint of habitat availability. According to the results of this research, this characteristic will tend to be maintained until 2050. However, its use with industrial approaches should consider that this species is distributed in geographical areas with habitat suitability conditions ranging from medium to low, suggesting that many of their populations may be climatically stressed and inappropriate use could put them at risk.

**Author Contributions:** J.L.B.-L. and R.R.-S., conceptualization; J.L.B.-L., U.R.-M. and J.S.B.-L., formal analysis; J.L.B.-L., J.L.E.-R., A.C., M.E., S.S.-E., C.M.R.-R., J.C.R.-S. and P.A.D.-M., investigation; J.L.B.-L., J.S.B.-L., M.E., and R.R.-S., writing – original draft; J.L.B.-L., R.R.-S., J.S.B.-L. and M.E., writing—review & editing; J.L.E.-R. and A.C., resources. All authors have read and agreed to the published version of the manuscript.

**Funding:** Thanks to the financial contribution of the Program for the Professional Development of the Government of Mexico, to the head of the Centre for Ecological Studies of the Faculty of Biological Sciences of Juárez University of the State of Durango, for the facilities provided for the accomplishment of this research, and to the reviewers who enriched the work with their valuable observations, and thanks for the collaboration of Richard Bledsoe for his help with the translation.

**Conflicts of Interest:** We declare the absence of any conflicts of interests at the time of submission.

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
