# Peer review of "Climatic Change and Habitat Availability for Three Sotol Species in México: A Vision towards Their Sustainable Use"

_sustainability, doi:10.3390/su12083455_

Round 1
Reviewer 1 Report
This paper has great potential and I enjoyed the topic and approach
Byline
Ln 5: Remove “Firstname Lastname”
Abstract
General comments: there is too much background information in the abstract and not enough about the novel findings of this study.
Ln 25: include genus and family name of plants at first mention and capitalize “sotol”
Ln 27: italicize “Dasylirion”
Ln 32: include naming authority of plant species at first mention
Keywords
Ln 37: “climatic” should be lowercase
Introduction
General comments: The last paragraph needs rewritten because the English is unclear, the aims of the study are vague, and the results (and their uses) aren’t well-defined. I am unsure what the authors are trying to say about habitat. The authors need to be clear about whether they are using an ecological niche model or species distribution model and use language that is clear and consistent throughout their paper so they aren’t confusing the two different types of models (see https://doi.org/10.1016/j.ecolmodel.2019.108837). If presence and use of the species is all that’s being evaluated, then why is climatic modeling needed? The authors need to be clear about the goals of their work. This section of the paper is rather brief and doesn’t adequately set up the justification for the study.
Ln 45: separate the ecological and cultural importance into two sentences
Ln 40: include genus and family name of plants at first mention
Ln 47: “are” should be “were”
Ln 53: Is “exponential growth” the correct description? The reader (and authors) cannot tell from only two data points.
Ln 57: This sentence is too long and complex – break it up.
Ln 64: This is unclear and needs rewritten – “In this regard, the habitat is sometimes characterized per se, as a distinguishable unit (e.g. 64 discrete vegetation units)…”
Ln 67: This is unclear and needs rewritten – “Under this criterion, and based on the aforementioned…”
Ln 70: How is the “greatest potential for exploitation” being defined and how does this work measure that and contribute to an understanding of that?
Ln 72: Sentence beginning with “It will, thus, contribute…” is unclear in its reference
Methodology
Ln 76: check font and start a new sentence after the parentheses
Ln 77: What is the source for the “known distribution?” Many species do not have their distribution completely mapped, so in your case how can you be sure the known distribution is accurate and the “erroneous and unreliable geographical records” are truly errors?
Ln 78: need naming authority at first mention of scientific names
Figure 1: Include a compass rose and scale bar. It doesn’t appear that each dot equals a single record from GBIF because I am unable in all three cases to count the same number of dots as records given on lines 78-79. Some dots are undoubtably covered by other dots. If the purpose of this figure is to provide the distribution area, and not each geographic observation, then it would be better to use a more standard species range map (e.g., https://en.wikipedia.org/wiki/Species_distribution#/media/File:Juniperus_communis_range_map.gif). The quality of the dots and text on the figure are low and are fuzzy around the edges.
Ln 86: What are the 19 environmental variables?
Ln 89: Need a reference that supports your 0.7 threshold for excluding correlated predictor variables.
Ln 93: Diurnal range of what? This needs described throughout the paper.
Ln 93-95: The abbreviations selected for the variables are uninformative and make this paper tedious because the reader must constantly refer to the key. Eliminate abbreviations altogether when possible (e.g., in the text, tables, Fig 5) and use informative abbreviations when necessary (e.g., in Fig. 2).
Results
General comments: PCA and AICc are only as good as the data that go into the models. Much more information is needed so the reader can evaluate what the PCA results mean and how strong the models are. For PCA, include the factor loadings because this indicates the contribution of each variable to each principal component and whether there is a positive and negative association. Discuss how the PCA analysis separates the variables, not simply the statistical results (which are relatively uninformative on their own). The data from AICc modeling needs to be provided, along with the r2 values so the reader can judge whether any model is decent. The table and figure headings all need more detail and abbreviations need defined. Tables and figures should be capable of standing on their own, and none of these could.
Table 1 and Figure 2 are relatively uninformative and the primary information can be included in the text only (“standard” is also misspelled in the Table).
Table 2 is misplaced.
Fig. 2: A table of factor loadings would give more detail and be quicker to interpret than this figure.
Ln 131: “greater association” is vague without describing whether it is a positive or negative association. This is also where the paper becomes especially tedious to read because the abbreviations used for the environmental variables are uninformative.
Ln 133: Okay, so the niche amplitudes differ along the two main PCs, but what does that mean in terms of associations with environmental variables?
Ln 174: This entire paragraph needs to state whether the variables are positively or negatively associated with the species distributions. The authors also revert back to the full names for the environmental variables, but use abbreviations in Fig 5, so the reader is forced to flip between three sections of the paper to evaluate the information (this paragraph, the figure, and the key for the abbreviations).
Ln 175: Commission and omission need explained.
Ln 186-218: It would be helpful to discuss not only where habitat is suitable or unsuitable, but also discuss more general observations about how suitability changes under the climate scenarios. E.g., Is habitat increasing or decreasing? Is there a general directional geographic shift in suitability or unsuitability? Some subtle changes are evident from looking at Fig 6, but it’s difficult to interpret much from that visual.
Ln 188: “Low” should be lowercase
Ln 192: Throughout the paper climate change scenarios are given by their numbers (i.e., 8.5, 2.6), but this is uninformative. What are the assumptions of these models? What do the models represent?
Fig 5 has spelling errors and a title error on the Y-axis.
Table 3 is uninformative and has misspellings.
Discussion
This section of the paper needs a major rewrite. The first paragraph is redundant with the results. Start with a broader overview of the significance of the results. There is verbiage throughout the discussion that would be better to move to the introduction to help establish background and justification for the study. The discussion needs to do a better job of explaining the significance of the results, applications, and impact of future climate on sotol and the industry and livelihoods that depend on it. The Discussion also needs to discuss the results and implications of other work and not the authors themselves (see my questions for Ln 252 as an example of items that need clarified). Much of the paper has minor writing errors that obscure what the authors are trying to say, but this is most problematic in the discussion. I find that I am not adequately able to evaluate some parts of this section because I don’t understand the authors.
Ln 229: this first sentence would be better placed in the introduction to expand on the rationale for this paper
Ln 237-242: this information would be better placed in the introduction
Ln 245: What is “high habitat?”
Ln 246: The writing in this paragraph needs work.
Ln 249: What is “low habitat” and “Under these evidences…?” Both sentences are unclear and need rewritten.
Ln 252: What does this mean: “…which coincides with that proposed by Vega-Cruz et al.?” And this, “On the other hand, our results reinforce what Brown, Johnson and Baltzer proposed?”
Ln 258: The writing in this paragraph needs work. “On the other hand…” is unnecessary since the first hand was never established. There are lots of other phrases in this paragraph that are similarly unnecessary. An easy way to identify whether the writing is straightforward and the wording is in the correct order is to look at comma use. Too many commas indicate clauses that are out of order and potentially unnecessary words.
Ln 272: “exploitation” appears to be used incorrectly here and throughout the paper
Author Response
The document has been reviewed by native English speakers
Ln 5: Remove “Firstname Lastname”
Reply: We have made the recommended changes
Abstract
General comments: there is too much background information in the abstract and not enough about the novel findings of this study.
Reply: We agree with the reviewer, we have eliminated information that may not be relevant to the summary and we have added information that gives a greater vision of our results.
Ln 25: include genus and family name of plants at first mention and capitalize “sotol”
Reply: We have made the changes suggested by the reviewer, we include the genus and family of the species under study:
The Industrial production and commercialization of distilled beverages from Sotol plants (Family: Asparagaceae, Subfamily: Zolinoideae and Genus: Dasylirion) has shown growth in recent decades; this condition involves a greater exploitation of the raw material that comes almost exclusively from natural populations, which could compromise the sustainability of the marginalized areas of northern Mexico.
Ln 27: italicize “Dasylirion”
Reply: We have revised the text and correctly spelled the scientific names
Ln 32: include naming authority of plant species at first mention
Reply: We have included the naming authority of the species under study: In the present work habitat availability was evaluated for the presence and use of the species Dasylirion wheeleri (S.Watson ex Rothr., 1878), Dasylirion cedrosanum (Trelease, 1911) and Dasylirion durangense (Trelease, 1911) in Mexico, considering different scenarios of climate change.
Keywords
Ln 37: “climatic” should be lowercase
Reply: We have cast in lowercase "climatic"
Introduction
General comments: The last paragraph needs rewritten because the English is unclear, the aims of the study are vague, and the results (and their uses) aren’t well-defined. I am unsure what the authors are trying to say about habitat. The authors need to be clear about whether they are using an ecological niche model or species distribution model and use language that is clear and consistent throughout their paper so they aren’t confusing the two different types of models (see https://doi.org/10.1016/j.ecolmodel.2019.108837). If presence and use of the species is all that’s being evaluated, then why is climatic modeling needed? The authors need to be clear about the goals of their work. This section of the paper is rather brief and doesn’t adequately set up the justification for the study.
Reply: The said paragraph has been rewritten as advised, mentioning therein briefly the aims and the outcomes of the study.
Ln 45: separate the ecological and cultural importance into two sentences
Reply: The ecological and cultural importance have been separately mentioned
Ln 40: include genus and family name of plants at first mention
Reply: The instruction complied for as advised.
Ln 47: “are” should be “were”
Reply: Corrected accordingly.
Ln 53: Is “exponential growth” the correct description? The reader (and authors) cannot tell from only two data points.
Reply: We agree with the reviewer and we have corrected the wording
Ln 57: This sentence is too long and complex – break it up.
Reply: The long sentenced has been fragmented .
Ln 64: This is unclear and needs rewritten – “In this regard, the habitat is sometimes characterized per se, as a distinguishable unit (e.g. 64 discrete vegetation units)…”
Reply: Rewritten as advised.
Ln 67: This is unclear and needs rewritten – “Under this criterion, and based on the aforementioned…”
Reply: Rewritten as advised.
Ln 70: How is the “greatest potential for exploitation” being defined and how does this work measure that and contribute to an understanding of that?
Reply: The sentence under mention has been replaced.
Ln 72: Sentence beginning with “It will, thus, contribute…” is unclear in its reference
Reply: The said sentence has been replaced.
Methodology
Ln 76: check font and start a new sentence after the parentheses
Reply: We believe that the way in which we have cited the GIF platform is correct. In some other works in which I have used data from this platform, the way in which we have cited it has been: (GBIF, http://www.gbif.org). See: https://journals.plos.org/plosone/article/file?id=10.1371/journal.pone.0185086&type=printable
Ln 77: What is the source for the “known distribution?” Many species do not have their distribution completely mapped, so in your case how can you be sure the known distribution is accurate and the “erroneous and unreliable geographical records” are truly errors?
Reply: We agree with the reviewer that it is necessary to add the frame of reference on which we rely to define possible erroneous geographical records. For this, the information provided by the National Institute of Forestry, Agricultural and Livestock Research (INIFAP) was taken as a reference, this information has been added to the document.
Ln 78: need naming authority at first mention of scientific names
Reply: This point was addressed in the summary and introduction section, naming authorities has been provided for each scientific name of the species under study.
Figure 1: Include a compass rose and scale bar. It doesn’t appear that each dot equals a single record from GBIF because I am unable in all three cases to count the same number of dots as records given on lines 78-79. Some dots are undoubtably covered by other dots. If the purpose of this figure is to provide the distribution area, and not each geographic observation, then it would be better to use a more standard species range map (e.g., https://en.wikipedia.org/wiki/Species_distribution#/media/File:Juniperus_communis_range_map.gif). The quality of the dots and text on the figure are low and are fuzzy around the edges.
We agree with the reviewer, figure 1 has been modified following the recommendations made by the reviewer.
Ln 86: What are the 19 environmental variables?
Reply: The name of each and every climate variable has been appended in the manuscript.
Ln 89: Need a reference that supports your 0.7 threshold for excluding correlated predictor variables.
Reply: We agreed with the reviewer and added the reference that supports why the value of 0.7 was considered to exclude predictor variables.
Ln 93: Diurnal range of what? This needs described throughout the paper.
Reply: The data have been provided which explains how the Mean Diurnal Range is obtained.
Ln 93-95: The abbreviations selected for the variables are uninformative and make this paper tedious because the reader must constantly refer to the key. Eliminate abbreviations altogether when possible (e.g., in the text, tables, Fig 5) and use informative abbreviations when necessary (e.g., in Fig. 2).
Reply: The reviewer's observation was addressed, abbreviations were removed from Table 2, Figure 5 and from some parts of the text.
Results
General comments: PCA and AICc are only as good as the data that go into the models. Much more information is needed so the reader can evaluate what the PCA results mean and how strong the models are. For PCA, include the factor loadings because this indicates the contribution of each variable to each principal component and whether there is a positive and negative association. Discuss how the PCA analysis separates the variables, not simply the statistical results (which are relatively uninformative on their own). The data from AICc modeling needs to be provided, along with the r2 values so the reader can judge whether any model is decent. The table and figure headings all need more detail and abbreviations need defined. Tables and figures should be capable of standing on their own, and none of these could.
Table 1 and Figure 2 are relatively uninformative and the primary information can be included in the text only (“standard” is also misspelled in the Table).
Reply: The information in Table 2 was included in the text and Figure 1 and Table 2 removed.
Fig. 2: A table of factor loadings would give more detail and be quicker to interpret than this figure.
Reply: We have addressed the reviewer's observation, we added the contribution values of the variables under study for components 1 and 2. This decision was made considering that, although the table with the values is important to understand more quickly, it, also a visual approach through the graph improves the understanding of this result significantly.
Ln 131: “greater association” is vague without describing whether it is a positive or negative association. This is also where the paper becomes especially tedious to read because the abbreviations used for the environmental variables are uninformative.
Reply: We have added the type of relationship (positive or negative) of the variables, as well as we have removed the abbreviations for the climatic variables.
Ln 133: Okay, so the niche amplitudes differ along the two main PCs, but what does that mean in terms of associations with environmental variables?
Reply: We consider that relating the width of the niches of PC 1 and PC 2 with the climatic variables leads us to interpret the results,
We have made the modification in the wording of the amplitudes of the niches, we have tried to better relate the amplitudes in PC 1 and PC2 with the environmental variables.
Ln 174: This entire paragraph needs to state whether the variables are positively or negatively associated with the species distributions. The authors also revert back to the full names for the environmental variables, but use abbreviations in Fig 5, so the reader is forced to flip between three sections of the paper to evaluate the information (this paragraph, the figure, and the key for the abbreviations).
Reply: the observations have been attended to, the paragraph was rewritten and the abbreviations were removed.
Ln 175: Commission and omission need explained.
Reply: We have indicated what the Commission and omission values refer to
Ln 186-218: It would be helpful to discuss not only where habitat is suitable or unsuitable, but also discuss more general observations about how suitability changes under the climate scenarios. E.g., Is habitat increasing or decreasing? Is there a general directional geographic shift in suitability or unsuitability? Some subtle changes are evident from looking at Fig 6, but it’s difficult to interpret much from that visual.
Reply: The variations in the habitat suitability have been mentioned in the discussion, we consider that in the results section it is not convenient to carry out an interpretation of the same.
Ln 188: “Low” should be lowercase
Reply: We have done the relevant corrections.
Ln 192: Throughout the paper climate change scenarios are given by their numbers (i.e., 8.5, 2.6), but this is uninformative. What are the assumptions of these models? What do the models represent?
Reply: We have added the W/m2 symbol to the CPR data, which indicates the average amount of solar energy absorbed per square meter on earth.
Fig 5 has spelling errors and a title error on the Y-axis.
We have done the relevant corrections
Table 3 is uninformative and has misspellings.
Table 3 has been removed.
Discussion
This section of the paper needs a major rewrite. The first paragraph is redundant with the results. Start with a broader overview of the significance of the results. There is verbiage throughout the discussion that would be better to move to the introduction to help establish background and justification for the study. The discussion needs to do a better job of explaining the significance of the results, applications, and impact of future climate on sotol and the industry and livelihoods that depend on it. The Discussion also needs to discuss the results and implications of other work and not the authors themselves (see my questions for Ln 252 as an example of items that need clarified). Much of the paper has minor writing errors that obscure what the authors are trying to say, but this is most problematic in the discussion. I find that I am not adequately able to evaluate some parts of this section because I don’t understand the authors.
Ln 229: this first sentence would be better placed in the introduction to expand on the rationale for this paper
Reply: We have done the relevant correction
Ln 237-242: this information would be better placed in the introduction
Reply: We have done the relevant correction
Ln 245: What is “high habitat?”
Reply: The introduction mentions what “high habitat” refers to: MaxEnt has demonstrated good predictability based on presence data and the habitat suitability is represented by a scale ranging from 0 (low suitability) to one (high suitability) (Elith et al. , 2006; Buckley et al. 2008; Navarro-Cerrillo et al., 2011, Jiménez-Valverde et al. 2011).
Ln 246: The writing in this paragraph needs work.
Reply: We have modified the phrase: The results obtained in the modeling for the year 2050 under conditions RCP 2.5 and RCP 8.5 show a slight increase in the habitat suitability for D. durangense and D. wheeleri. However, for D. cedrosanum, only areas with low habitat suitability are presented.
Ln 249: What is “low habitat” and “Under these evidences…?” Both sentences are unclear and need rewritten.
Reply: The sentences have been rephrased.:
Ln 252: What does this mean: “…which coincides with that proposed by Vega-Cruz et al.?” And this, “On the other hand, our results reinforce what Brown, Johnson and Baltzer proposed?”
Reply: The sentence has been corrected:
Therefore, it is possible to point out that, although the future climate trend is framed by an increase in CO2 and temperature [37], this condition cannot be correlated in general with an increase in the habitat suitability of the species of the genus Dasylirion. On the other hand, our results reinforce the hypothesis that species with narrow niches are more susceptible to climate change. Since in this study the species that presented the smallest climatic niche (D. cedrosanum), showed the greatest decrease in the suitability of habitat considering the increase in in atmospheric CO2 concentrations.
Ln 258: The writing in this paragraph needs work. “On the other hand…” is unnecessary since the first hand was never established. There are lots of other phrases in this paragraph that are similarly unnecessary. An easy way to identify whether the writing is straightforward and the wording is in the correct order is to look at comma use. Too many commas indicate clauses that are out of order and potentially unnecessary words.
Reply: We have modified the paragraph trying to understand it better.
Ln 272: “exploitation” appears to be used incorrectly here and throughout the paper
Reply: We have replaced the word exploitation with utilization throughout the paper.

Reviewer 2 Report
I do think the research topic is interesting. Perhaps the design and, consequently, the methods may be improved. I would suggest a minimum of fieldwork. Unless the aim of the research is redefined to present it as a first and explorative phase that will implement, in a second phase and at more detailed scales, the fieldwork necessary to check the reliability and shortcomings of the databases used.
Author Response
Reply: At the moment we are unable to carry out field work due to the COVID 19 environmental contingency, therefore we have based the reliability of the data on our field experience, as well as statistics obtained from the area under the curve. Various studies use the AUC as a reliable source to validate niche models.
Reference:
https://doi.org/10.1111/j.1365-2664.2008.01516.x
https://doi.org/10.1371/journal.pone.0185086
https://link.springer.com/article/10.1007/s11284-015-1318-7

Round 2
Reviewer 1 Report
This paper is greatly improved and commend the authors for the major revisions they made. However, there are still necessary revisions. Chief among the needed revisions is correcting the English language and writing structure. Some of these problems are minor and will likely be detected by the journal’s editorial team, but there are numerous errors throughout that still obscure what the authors intend to say. I pointed many of these out during the first review and doing so in this draft would take great effort. I know the authors had native English speakers review the paper, but the changes aren’t enough. For example, the abstract has capitalization, spacing, and punctuation (semi-colon) errors. There is odd phrasing throughout (e.g., “species on trial”) and sometimes incorrect phrases (e.g., instead of “niche's spatial analysis from a correlative approach,” say “an ecological niche model”). These examples are specific to the abstract, but could be said about all parts of the paper.
This isn’t an exhaustive list, but some other changes that are needed (some of which I mentioned during the first review).
Byline
Ln 5: Remove “Firstname Lastname”
Abstract
“Utilization” should be “use” throughout this ms and it’s not useful as a key word. Utilization is overused and used incorrectly in the English language.
Introduction
Ln 67: habitat and environment aren’t synonyms
Ln 73: remove “niche's spatial analysis” and refer to what it really is (i.e., “ecological niche modeling”)
Ln 81: Sentence beginning this paragraph is unclear
Methodology
Ln 98: The authors must not have understood my meaning during the first review. “Mean Diurnal Range” is uninformative. Yes, they now include how it is calculated which means they are discussing temperature, so this environmental variable should be named “Mean Diurnal Temperature Range,” otherwise it could mean light, precipitation, or a host of other environmental variables.
Also, unless the “bio” abbreviations are used elsewhere in the paper, including all 19 of them in parentheses after the full name of the environmental variables is unnecessary
Results
I still see no data from AICc modeling, which needs to be provided, along with the r2 values so the reader can judge whether any model is decent.
The table and figure headings are all improved, but still need to be much more specific in nearly all cases. Tables and figures should be capable of standing on their own, and most of these could not. Table 2 especially does not make sense on its own and needs more description. There are also Spanish spellings in this table.
Discussion
This is the other section of the paper where writing is still especially problematic, beginning with the very first sentence (which makes no sense to me no matter how many times I reread it and try to interpret it). Habitat is specific to an individual or species, not a community of species. In the second sentence, “An analysis based on the management of PCA” doesn’t make sense. I could go on with other examples of problems, but as I previously mentioned, that would take a lot of my time due to their abundance.
Ln 227: Species don’t occupy different types of habitat. Everything they occupy is their habitat.
Author Response
Ln 5: Remove “Firstname Lastname”
Reply: The names “Jorge Luis Becerra-López” and “Jesús Salvador Becerra-López” refer to two different researchers (Co-authors), it is not possible to carry out the deletion suggested by the reviewer.
Abstract
“Utilization” should be “use” throughout this ms and it’s not useful as a key word. Utilization is overused and used incorrectly in the English language.
Reply: We have changed the word utlization to use. Also, we removed the utlization keyword and it has been replaced by niche.
Ln 67: habitat and environment aren’t synonyms
Reply: We have corrected the wording, the word habitat has been removed.
Ln 73: remove “niche's spatial analysis” and refer to what it really is (i.e., “ecological niche modeling”)
Reply: We have replaced "niche's spatial analysis" with "ecological niche modeling (see:Ln 77)".
Ln 81: Sentence beginning this paragraph is unclear
Reply: We have re-written the sentence under question.
Methodology
Ln 98: The authors must not have understood my meaning during the first review. “Mean Diurnal Range” is uninformative. Yes, they now include how it is calculated which means they are discussing temperature, so this environmental variable should be named “Mean Diurnal Temperature Range,” otherwise it could mean light, precipitation, or a host of other environmental variables.
Reply: We have changed the name of the variable “Mean Diurnal Range” to “Mean Diurnal Temperature Range”, according to the reviewer’s suggestion.
Revisor1: Also, unless the “bio” abbreviations are used elsewhere in the paper, including all 19 of them in parentheses after the full name of the environmental variables is unnecessary.
Reply: The abbreviations (bio) are used in the figure 2, for this reason we cannot remove them from the text.
Results
I still see no data from AICc modeling, which needs to be provided, along with the r2 values so the reader can judge whether any model is decent.
Reply: We consider that adding a table with the AIC values for each model would be somewhat tedious for the reader, since for each species 160 models were generated during the calibration process and out of these the one with the lowest AIC was chosen. In this case we selected the model that obtained an AIC of "0". However, we agree with the reviewer that the AIC value that was considered for the selection of the best calibration should be indicated. In response to this, we have added the following paragraph in the results section:
From the calibrations of the models for the species D. cedrosanum, D. durangense and D. wheeleri, the one that obtained a value of 0 was selected. This under the criterion that the candidate calibrations must be classified according to their AIC and the calibration that presents the minimum AIC will be the best.
Regarding the R2 data, we consider that it is not convenient to append the correlation tables since for each species there are 361 possible combinations, which can be cumbersome for the reader. That’s why, we only mentioned that those variables that presented a correlation equal to or greater than 0.7 were eliminated. Likewise, the importance of each of the 19 climatic variables in the ecology of the three species under study was considered.
The table and figure headings are all improved, but still need to be much more specific in nearly all cases. Tables and figures should be capable of standing on their own, and most of these could not. Table 2 especially does not make sense on its own and needs more description. There are also Spanish spellings in this table.
Reply: we have attended the reviewer's observation.